# GRAPHICAL-TS: AN INTERACTIVE AI PIPELINE FOR MULTIVARIATE TIME SERIES WITH GROUND-TRUTH GRAPHICAL MODELING

## ABSTRACT

We present `Graphical-TS`, an interactive simulation framework for multivariate time series (MTS) incorporating spatiotemporal causal graphical models. The system offers extensive customizability, enabling users to define and modify causal dynamics with uncertainty in spatiotemporal relationships and functional mappings. `Graphical-TS` integrates expert knowledge, supports MTS simulation, and allows for the fusion of real-world MTS data, facilitating a dynamic interplay between data-driven learning and domain expertise. The system iteratively enhances causal relationships and simulated data by simulating MTS data based on specified causal graphs, performing causal discovery from real or simulated MTS, and enabling the integration and refinement of expert knowledge with learned causality. This approach progressively enhances the quality of both the causal models and the data they produce, facilitating tasks such as time series forecasting and imputation. By customizing functional mappings, scenario-driven distribution shifts can be modeled, enabling robust testing of time series algorithms. We compared state-of-the-art causal discovery methods on datasets generated by `Graphical-TS`. The empirical results demonstrate the platform's consistent performance compared to existing methods while offering versatility under distinct scenarios. This enables users to explore datasets more thoroughly and drive improvements in causal discovery research. With an intuitive user interface that connects domain experts and algorithm developers, `Graphical-TS` empowers users to manipulate causal relationships, embedding domain knowledge into machine learning workflows. Originally developed to study physiological dynamics in patients, the system has broad applicability across various fields, offering a versatile platform for generating MTS datasets with known dynamics, validating causal discovery algorithms, and advancing research in time series analysis.

## 1 INTRODUCTION

Multivariate time series (MTS) data plays a crucial role across various domains such as health, transportation, earth science, and finance. A key aspect of MTS is the inter-correlation among variables, which helps model the underlying mechanisms generating multivariate signals. These inter-relationships can be represented as causal graphs that map the influence of one variable on another without forming loops—formally expressing causality. For example, in healthcare, graphical models have been used to improve medical diagnosis and prognosis, as seen in studies like predicting sepsis onset using machine-learned causal probabilistic networks based on electronic health records data (Valik et al., 2023) and describing functioning in people living with spinal cord injury across 22 countries (Ehrmann et al., 2020). Additionally, graphical models have been applied to infer cellular networks using probabilistic graphical models (Friedman, 2004) and to detect and quantify causal associations in large nonlinear time series datasets (Runge et al., 2019a). These models can capture complex causal dependencies in various domains, offering insights into underlying processes and supporting decision-making.

In some cases, like monitoring a circuit board where pin voltages are observed, the circuit's structure provides a clear representation of the underlying graph generating these signals. However, in most real-world situations, the causal graph is at least partially unknown. Expert knowledge can help elicit the causal graph; for instance, the selection of sensors in the signal collection often reflects

prior assumptions about causal relationships. While this may introduce bias, incorporating expert knowledge remains valuable as long as it is sufficiently reliable.

Additionally, data-driven methods known as causal discovery—comprising constraint-based, functional-based, and score-based approaches—are used to learn the graph from data. Causal discovery is particularly important in time series analysis, where temporal dependencies add an extra dimension of complexity. In this context, expert knowledge often serves as a valuable foundation for building graphical models, providing a starting point that can be iteratively refined through a combination of human input and algorithmic learning.

## 1.1 NOTATIONS

We denote the set of features by $X^j$, where $j \in [N]$ and $N$ is the number of features. Each feature $X^j$ also represents the associated random process $X^j : t \mapsto X_t^j$, mapping time steps to random variables. Following standard conventions, we use $X_t^j$ to denote the random variable at time step $t$, and $x_t^j$ for its observed value (realization).

In our graphical model, the nodes correspond to the random variables $X_t^j$. Since there is no ambiguity, we use $X_t^j$ to refer both to the random variable and the corresponding node in the graph. The set $\mathrm{Pa}(X_t^j)$ denotes the causal parents of $X_t^j$ within the graph. Extending this notation to realizations, $\mathrm{Pa}(x_t^j)$ represents the collection of observed values corresponding to the parents of $X_t^j$.

For edge notation, we use $(X, Y)$ to represent a directed edge from $X$ to $Y$, indicating that $X$ is a direct cause of $Y$. An undirected edge between $X$ and $Y$ is denoted by $\{X, Y\}$.

## 2 RELATED WORK

Researchers frequently generate multivariate time series (MTS) data for evaluating algorithms, as readily available datasets are scarce. Simulating MTS with predefined dynamics is a common practice that allows researchers greater control over experimental conditions and the evaluation process. However, we observe a significant redundancy in these data generation efforts, particularly in crafting graphical models and specifying functional relationships. We contend that domain experts are best suited to define and identify the causal relationships between variables. At the same time, algorithm developers should concentrate on refining the algorithms themselves, rather than investing time in coding simulations for each model they develop. One of the primary objectives of this paper is to bridge the gap between domain experts and algorithm developers, thereby streamlining the research process and enhancing the efficiency of causal discovery studies.

### 2.1 GENERATION OF MULTIVARIATE TIME SERIES WITH GRAPHICAL MODEL KNOWLEDGE

**CausalTime**(Cheng et al., 2024) is a pipeline that generates synthetic time series data by first fitting a nonlinear autoregressive model to real-world data to infer causal relationships, which are then used to construct a causal graph for data generation. While CausalTime employs discriminative scores to ensure generated data matches real-world distributions, this reliance on aggregate metrics can mask important local patterns and nuances, especially in complex domains like healthcare where human expertise is crucial. While CausalTime's use of discriminative scores provides a valuable quantitative approach to ensuring data quality, `Graphical-TS` complements this by incorporating visual analysis tools that enable experts to directly identify and assess discrepancies between synthetic and real data. This human-in-the-loop approach works in conjunction with CausalTime's metric optimization approach, combining the benefits of quantitative optimization with expert-guided refinement. Table 1 summarizes the key differences between these two approaches.

### 2.2 CAUSAL DISCOVERY FOR MULTIVARIATE TIME SERIES DATA

Causal discovery in multivariate time series (MTS) data focuses on uncovering temporal and contemporaneous causal relationships among variables that evolve over time. Various methods have been developed for this purpose, each falling into different methodological categories.

**PCMCI** (Runge et al., 2019b) is an algorithm designed to efficiently discover causal links in high-dimensional time series data. It begins by applying conditional independence (CI) tests to filter out irrelevant variables, thereby reducing dimensionality. PCMCI enhances the selection of conditioning sets and distinguishes between lagged and contemporaneous dependencies, which improves its ability to detect causal relationships in the presence of autocorrelation and confounders. Variants such as PCMCI+(Runge, 2020) and LPCMCI(Gerhardus & Runge, 2020) have been introduced to

| | Graphical-TS | CausalTime |
|---|:---:|:---:|
| Experts' Engagement | ✓ | ✗[1] |
| Functional Definition | ✓ | ✗ |
| Realistic | ✗[2] | ✓ |
| Fusion with Real-world Data | ✓[3] | ✗ |

Table 1: Feature comparison between Graphical-TS and CausalTime. Notes: 1) While CausalTime can incorporate expert knowledge as a prior graph, this input becomes diluted through the pipeline. 2)&3) Via the human-in-a-loop iteration, Graphical-TS allows the continuous refinement through real-world data fusion to increase the realism.

further refine conditioning set selection and handle latent confounders, providing greater flexibility and accuracy across various data scenarios.

**VarLiNGAM** (Hyvärinen et al., 2010) combines vector autoregressive (VAR) models with linear non-Gaussian acyclic models (LiNGAM) to infer causal structures from time series data. By leveraging non-Gaussianity and the autoregressive nature of time series, VarLiNGAM estimates both instantaneous and lagged causal effects without requiring prior knowledge of the network structure. This method is adept at handling complex systems where variables influence each other over time, particularly when the data exhibit non-Gaussian distributions.

**DYNOTEARS** (Pamfil et al., 2020) extends the NOTEARS (Zheng et al., 2018) framework to address dynamic time series data. It estimates both contemporaneous (intra-slice) and time-lagged (inter-slice) causal relationships by formulating causal discovery as a continuous optimization problem. The method imposes an acyclicity constraint to ensure a directed acyclic graph (DAG) and assumes that the network structure remains static over time. Notably, DYNOTEARS is scalable to high-dimensional datasets and is implemented in the CausalNex library, facilitating its application in a wide range of time series analyses.

## 3 GRAPHICAL-TS: A PYTHON LIBRARY FOR EXPERT KNOWLEDGE INTEGRATION AND TIME SERIES DATA GENERATION

In this section, we will first introduce the mathematical model of the graphical dynamical system along with assumptions. Following that, we will explain the implementation details.

### 3.1 THE MATHEMATICAL MODEL

The mathematical model we use is called the *discrete-time structural causal process* (Runge, 2020) (DSCP), which we will define in 3.1.

**Definition 3.1** (Discrete-time structural causal process). A *discrete-time structural causal process* is a multivariate dynamical system determined by two components:

1. A vector process of of $N$ variables $\mathbf{X}_t = (X_t^1, \ldots, X_t^N)$

2. A set of mappings $X_t^j = f^j\left(\mathrm{Pa}(X_t^j), \eta_t^j\right), \quad j = 1, \cdots, N$

where $\mathrm{Pa}(X_t^j) = \{X \in \mathbf{X} \mid \mathbf{X} \in \{\mathbf{X}_t, \mathbf{X}_{t-1}, \ldots\}\} \setminus X_t^j$, and $\eta_t^j$ is a random variable from an uncertainty process $\eta^j$

*Remark.*

- In other parts of this paper, we will refer to the set $\boldsymbol{f} = \{f^j \mid j \in [N]\}$ as the functional form of the process

- When not specified otherwise, "parents" indicates the union of the upstream nodes from both the contemporaneous and lagged edges

**Definition 3.2.** We call an edge $(X_{t-\tau}^i, X_t^j)$ to be

- *contemporaneous* if $\tau = 0$

- *lagged* if $\tau > 0$

The DSCP defined is node-oriented given the time and index of a variable, as depicted in figure 1. Rolling through time the generation dynamics illustrates a spatial-temporal process whose realization is a MTS. The spatial perspective is due to the mappings across different variables despite their progress in time, and the temporal perspective is due to the existence of lagged edges. Taking clinical monitoring as an example, $\mathbf{X}_t = (X_t^1, \dots, X_t^N)$ could record the collection of values of human temperature, ECG(electrocardiogram), blood pressure, and so on, at the timestep $t$.

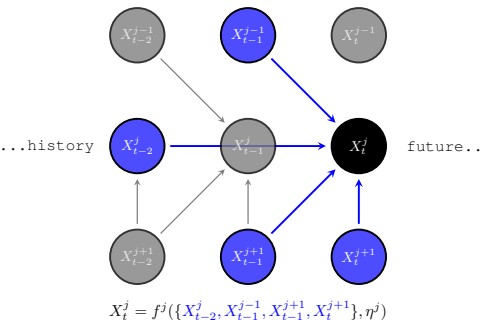

$$X_t^j = f^j(\{X_{t-2}^j, X_{t-1}^{j-1}, X_{t-1}^{j+1}, X_t^{j+1}\}, \eta^j)$$

Figure 1: The process is oriented by the computation of the value of a node at a time unit. The explicit mapping from a parent node to its child nodes should be defined for the simulation. Generally, a node can have multiple parent nodes, therefore the effects of those parents can be considered together or separately. Ideally, users should collectively specify this N-to-1 mapping function for each node. While sometimes it is hard to achieve, we use a default mapping for each edge if it is not specified. In addition, we enable the user to define mappings that correspond to non-intersecting subsets of the parents of a node. In this case, the effects from different parent groups will be added.

Depending on domain research, the function $\boldsymbol{f}$ can be complicated. The nature of $\boldsymbol{f}$ varies widely across different fields, reflecting a broad spectrum of complexity and characteristics. In some domains, $\boldsymbol{f}$ might be simple and linear, while in others, it could be highly nonlinear and intricate. This diversity makes it challenging to emphasize any specific form of $\boldsymbol{f}$ as universally representative. Consequently, my paper focuses not on a particular form of $\boldsymbol{f}$, but rather on the underlying principles and applications that transcend specific functional forms. This approach acknowledges the vast differences in how $\boldsymbol{f}$ can manifest depending on the research context.

### 3.2 ASSUMPTIONS

**Uncertainty Processes**. For simplification, we assume the functional $f^j$ are additive with respect to $\mathrm{Pa}(X_t^j)$ and the uncertainty process $\eta_t^j$, which means $X_t^j = f^j\left(\mathrm{Pa}(X_t^j), \eta_t^j\right) = f^j\left(\mathrm{Pa}(X_t^j)\right) + f^j\left(\eta_t^j\right)$. Noticing that $f^j\left(\eta_t^j\right)$ is a new uncertainty process, we can further define $g_t^j = f^j\left(\eta_t^j\right)$. As a result, we land on the model:

$$X_t^j = f^j\left(\mathrm{Pa}(X_t^j)\right) + g_t^j, \quad j = 1, \cdots, N$$

**Additive Functional Edges**. The edge function is a mapping from multiple nodes to their common child, i.e. an N-to-1 mapping. It is not always realistic to specify the joint casual effect in a single attempt. Usually it is more natural to progressively input 1-to-1 mappings, or k-to-1 mappings, where k is a considerably small number than N. Therefore we assume $f^j$ is additive with respect to the local causal effects by a subset of parents. To formally model this, we first partition $\mathrm{Pa}(X_t^j)$:

$$\mathrm{Pa}(X_t^j) = \bigcup_{k=1}^m \mathrm{Pa}_k(X_t^j)$$

where $m \le N$ and $N = |\mathrm{Pa}(X_t^j)|$. In practice, the partition depends on the real-world human interaction. Then, we define the local causal effects as:

$$f^j{}_k : \mathrm{Pa}_k(X_t^j) \mapsto X_t^j$$

Finally, the additive assumption can be written as:

$$f^j(\cdot) = \sum_{k=1}^m f^j{}_k(\cdot)$$

It is useful to assume $f^j{}_k$ to be non-linear for all $k$ because otherwise we can decompose $f^j$ further with a partition of smaller granularity. Additionally, one may notice that when $m = 1$, it is equivalent to possess the N-to-1 joint functionals.

### 3.3 Implementation

In this section, we will introduce how we programmatically construct the process in the definition 3.1.

**a). Graphical Model**
The first component is the causal graph which defines the spatial-temporal process itself. A causal graph is legitimate when it is a directed acyclic graph (DAG), which means it contains no loop. To achieve this, avoiding loops in the contemporaneous edges is necessary, as the lagged edges can only broadcast effect towards the future.

The implementation of the graphical model is built upon the NetworkX (Hagberg et al., 2008) library, a popular Python package in network analysis.

**b). Functional Mappings**
Once the graphical model is defined, a naive mapping is assigned to every edge. The naive mapping can either be an identical mapping, which copies the same value to the successor, or a null mapping which does nothing to the successor.

The placeholder mappings tends to create unstable MTS unless the human knowledge starts to be incorporated. With the additive functional edges assumption mentioned in 3.2. The human experts will incrementally improve those edges. We provide several options:

The following options are available for constructing functional maps, each offering distinct advantages and applications:

i. *Parameterized Templates*: These templates provide flexible multivariate mappings between categorical, continuous, and binary variables through customizable parameters. Their primary advantages include rapid implementation and agile adaptation to changing requirements, enabling efficient model prototyping and refinement.

ii. *Loaded Models*: We enable the deployment of more sophisticated models that employ machine learning algorithms or research-validated equations to transform parent nodes into succeeding nodes. This approach offers several key benefits: it maintains high reliability through data-driven validation, leverages established computational techniques, and facilitates direct comparison between expert knowledge and empirical models, enabling users to evaluate how their domain expertise aligns with quantitative findings.

### 3.4 Edge Functions with Expert Knowledge

The main functional form that is used to incorporate expert knowledge. On the user interface, experts will quantify a link by specifying the scale of the effect. The absolute value of the scale indicates how much the downstream variable is affected by its parent. A positive scale indicates increase/decrease will broadcast to the child node and vice versa.

The expert edge is different depending on the combination of data types of the parent and child. When a continuous variable is directed to a binary or categorical variable, we group the values in a set of intervals so they fall into indexed categories. When a binary variable is directed to a continuous variable, a s

### 3.5 Perturbation of the Graphical Model

`Graphical-TS` introduces distribution shifts by allowing for the modification of functional mappings. It enables users to simulate the impact of various scenarios on the graphical model. For instance, in a clinical setting, the physiological dynamics of patients can differ significantly between an ICU environment and routine care. Notably, the scenario itself can be thought of as a time-series variable that evolves infrequently. Additionally, `Graphical-TS` accounts for structural perturbations based on a globally accepted graph, which is considered to represent collective knowledge. Perturbations can occur in terms of both the connectivity between variables and the temporal lags involved. We offer APIs and an interactive interface to facilitate these structural adjustments, enabling users to explore scenario-driven distribution shifts within the functional map.

## 4 USER INTERFACE FOR EXPERT KNOWLEDGE INTEGRATION

To facilitate the process of creating, editing, and refining causal graphs, we have developed a user interface, which we show in figure 2. This interface is designed to support researchers and practitioners in the iterative process of causal discovery and model refinement.

The user interface is a comprehensive tool that enables users to construct, visualize, and interact with graphical models for time series data. The interface enhances the efficiency and effectiveness of causal discovery by providing a robust environment for both novice and experienced users. A more comprehensive introduction of the interface is provided in the Appendix A.

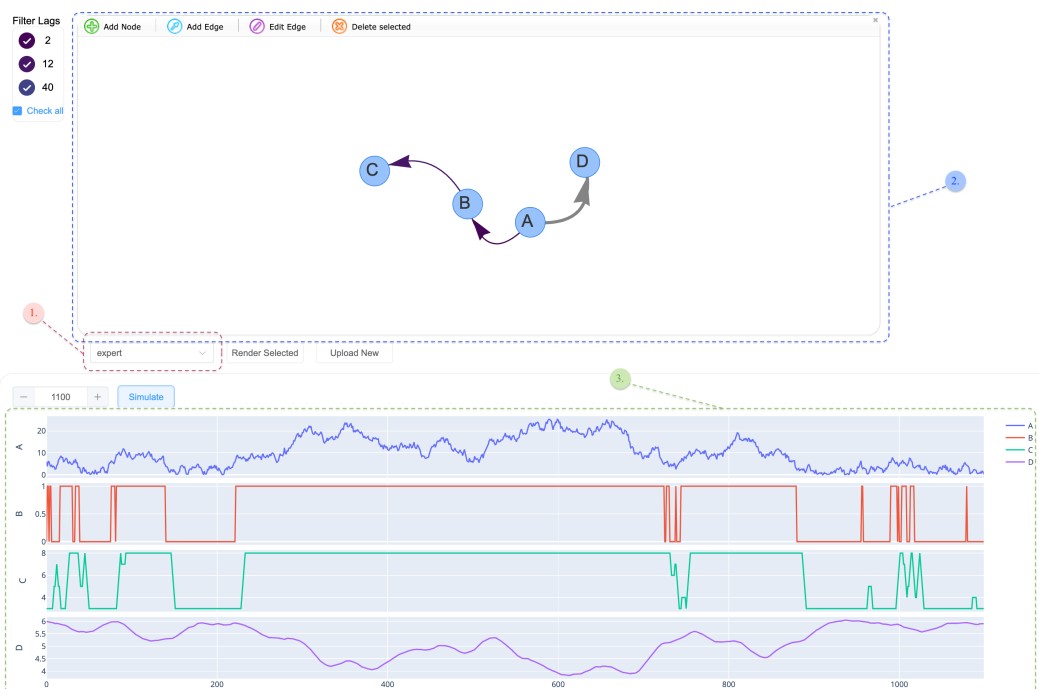

Figure 2: The user interface for expert knowledge input. The main view is designed to display and edit the graphical model and the generated MTS. The drop-down list 1 is for loading different graphical models. The image shows a graph that is created from scratch by an expert, but in practice, it is possible to load graphs that come from causal discovery algorithms in another workflow. Panel 2 is the area where the user can create and edit a graph either loaded or fresh-created. Adding a node is intuitive clicking the empty area which triggers a dialogue for confirmation, which we will show in figure 6. Dragging from one node to a different node will create a temporary edge which also requires confirmation from a dialogue. Area 3 is an area to preview the generated MTS given a period defined by several time steps, which is 1100 in this example.

### 4.1 THE ASSUMED ROLE OF HUMAN EXPERTS

The interface of `Graphical-TS` functions as a labeling system that integrates human insights, encompassing both theoretical knowledge from idealized or controlled research and empirical knowledge gained from real-world experience, where conditions are often unpredictable and data is noisy. For example, in a clinical environment, experts might use `Graphical-TS` to record their observations on how physiological parameters like heart rate, blood pressure, and respiratory rate interact under everyday conditions. These interactions are influenced by various unpredictable factors such as stress, medication, or underlying health conditions. Unlike data from controlled laboratory experiments with isolated variables, this real-world experience offers a more nuanced understanding of parameter behavior in the complex context of patient care. This empirical knowledge, integrated through Graphical-TS, refines the model to better reflect actual clinical dynamics. The system as-

sumes users are domain experts providing valuable insights based on their experience. While individual input may be uncertain or biased, the system values the collective knowledge of multiple experts, recognizing that human opinions are inherently imperfect. To address these imperfections, the graphical model is iteratively improved through a collaborative process that combines human input with algorithmic adjustments, ensuring a more accurate representation of the underlying causal structures by balancing expert knowledge with data-driven refinements.

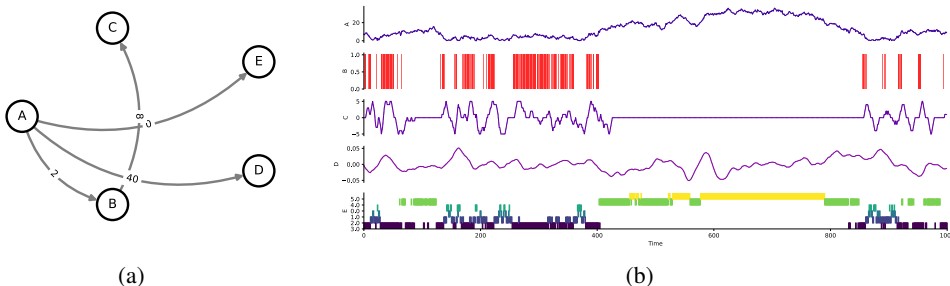

(a)                                              (b)

Figure 3: **(a)** Graphical model created by an expert interacting with the user interface, representing causal relationships between mixed-type variables, including continuous, binary, and categorical variables. The edges indicate the direction and lag of causal influences among the variables. **(b)** Simulated time series generated based on the graphical model, showing the dynamics of the mixed-type variables. Variable B is binary (red), and variable E is categorical (multi-colored segments), while the remaining variables exhibit continuous values. The time series captures the causal dependencies as defined by the expert-generated model.

## 5 BENCHMARKING CASUAL DISCOVERY ON GRAPHICAL-TS

A common practice in causal discovery research is to generate Erdős–Rényi (Erdos & Rényi, 2022) graphs or hand-crafted graphs with simple functional mappings (Du, 2023), a step typically performed by algorithmic researchers. However, because causal discovery is highly domain-specific, it is important to involve domain experts in the data generation process. Although human labeling may introduce biases, it captures important empirical aspects that statistical methods cannot. Furthermore, in many scenarios, such as with physiological dynamics data, the original time series records are highly privacy-sensitive. Graphical models labeled by experts serve as summaries and desensitized representations of real-world multivariate time series (MTS). These graphs can be used to generate time series that recover aspects of the real-world MTS while being safer to publish as benchmarks.

### 5.1 CAUSAL DISCOVERY FOR TIME SERIES DATA

As mentioned in section 2, there is a zoo of causal discovery algorithms, including constraint-based methods and gradient-based methods. There is no fixed quantitative scheme for measuring the quality of a learned graph. For constraint-based methods and Granger Causality-based methods, edges are added by statistical tests; in other words, they are measured by nature. In gradient-based methods, a graph is learned by reinforcing the acyclicity. Due to the difference in their assumptions, the resulting causal model might not align. Although researchers can choose what algorithms they want, with the presence of ground truth, the algorithm that better reconstructs the ground truth graph on which the simulation runs should be preferred. In a scenario of expert knowledge incorporation, the edge relationship input by an expert should be learned, which means the algorithms learn how humans decide whether an edge should be added or modified. As we will see in the following section, the simulated data of GRAPHICAL-TS will be submitted to both kinds of methodology, and evaluated by comparing it to the ground truth.

**Metrics**   We evaluate the performance of causal discovery algorithms using the **F1 Score**, which provides a balanced measure of precision and recall, offering a comprehensive assessment of accu-

racy. The F1 Score is defined as:

$$\text{F1 Score} = 2 \times \frac{\text{Precision} \times \text{Recall}}{\text{Precision} + \text{Recall}},$$

where Precision $= \frac{|\text{TP}|}{|\text{TP}|+|\text{FP}|}$ and Recall $= \frac{|\text{TP}|}{|\text{TP}|+|\text{FN}|}$. In our experiment, true positives (**TP**) refer to cases where both the source and target nodes of an edge are correctly identified, and the lag falls within an acceptable window. False positives (**FP**) are incorrectly identified edges, while false negatives (**FN**) represent missed causal edges. We consider an edge is correctly identified when a). both the source node and the target node are correct and b). the lag calculated falls within a window of the correct lag.

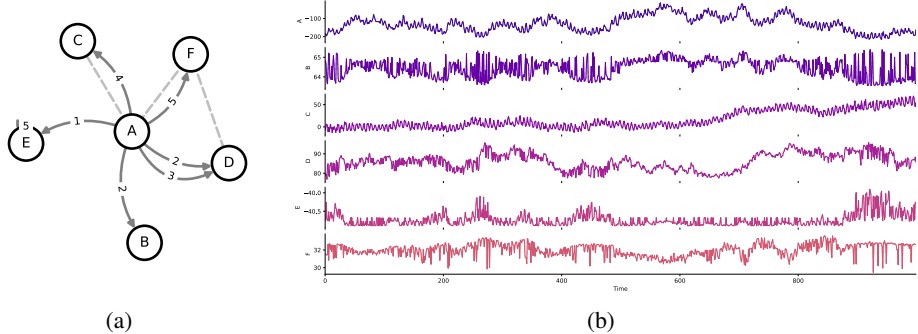

(a)  (b)

Figure 4: **(a)** Simulated graphical model representing the causal structure of six observed variables (A-F), where the dashed arrows indicate the presence of a latent confounder influencing the system. No contemporaneous links are included in the model. **(b)** The corresponding simulated multivariate time series generated using a 2-layer multilayer perceptron (MLP) with ReLU activation functions.

**Simulated Datasets**  We developed multiple simulated datasets featuring graph instances with a consistently fixed number of nodes to *five* to allow for an in-depth exploration of complex structural types while maintaining computational manageability. The dataset is systematically varied along three major dimensions:

i). *Structural Configurations.* The structural configurations chosen: *tree*, *star*, and *cycle*, represent fundamental topologies that offer distinct challenges in causal inference and reflect common patterns observed in real-world networks.

ii). *Maximum Lag of Edges.* This influences the temporal depth of causal connections, allowing us to examine how well algorithms can untangle delayed influences that are typical in dynamic systems. For this setting, we fixed the structure of the edge-node ratios of the structure.

iii). *Edge-node Ratios.* The edge-node ratio, including all lagged edges, is varied to test the resilience of causal discovery algorithms under varying degrees of connectivity and information density. For this setting, we fixed the maximum lag of edges.

We only considered lagged graphs in our simulation. This allows us to use cycles because lag in edges prevents cyclicity. While it is possible to input contemporaneous effect for `Graphical-TS`, we consider it a special case of lag effect with lower sampling frequency. For the functional maps, random multi-layer perceptrons (MLP) are employed. We note here the generated data can also as tabular data by randomizing the temporal ordering to disrupt autoregressive dependencies, which will be discussed later in section 5.2

**Results**  With the aforementioned setup, the benchmarking results we get are demonstrated in figure 5, together with 7 and 8 in appendix B

## 5.2 Causal Discovery for Tabular Data

The simulated data can also be used for causal discovery for tabular data. We can notice that discrete-time structural causal process(DSCP) 3.1 can be degenerated into a casual graph without lagged edge and the simulated data can be used as tabular data when permutation is performed for a long time.

## 6 TIME SERIES FORECASTING AND IMPUTATION

Synthesis of multivariate data has been widely used for developing and evaluating algorithms in time series forecasting and imputation, especially when real-world datasets are limited. Traditionally, this data generation has relied on basic causal models combined with randomly structured graphs and short-term lagged effects to approximate variable interactions over time. While these approaches have proven useful, they may not fully capture the complex spatial and temporal dependencies found in real-world systems.

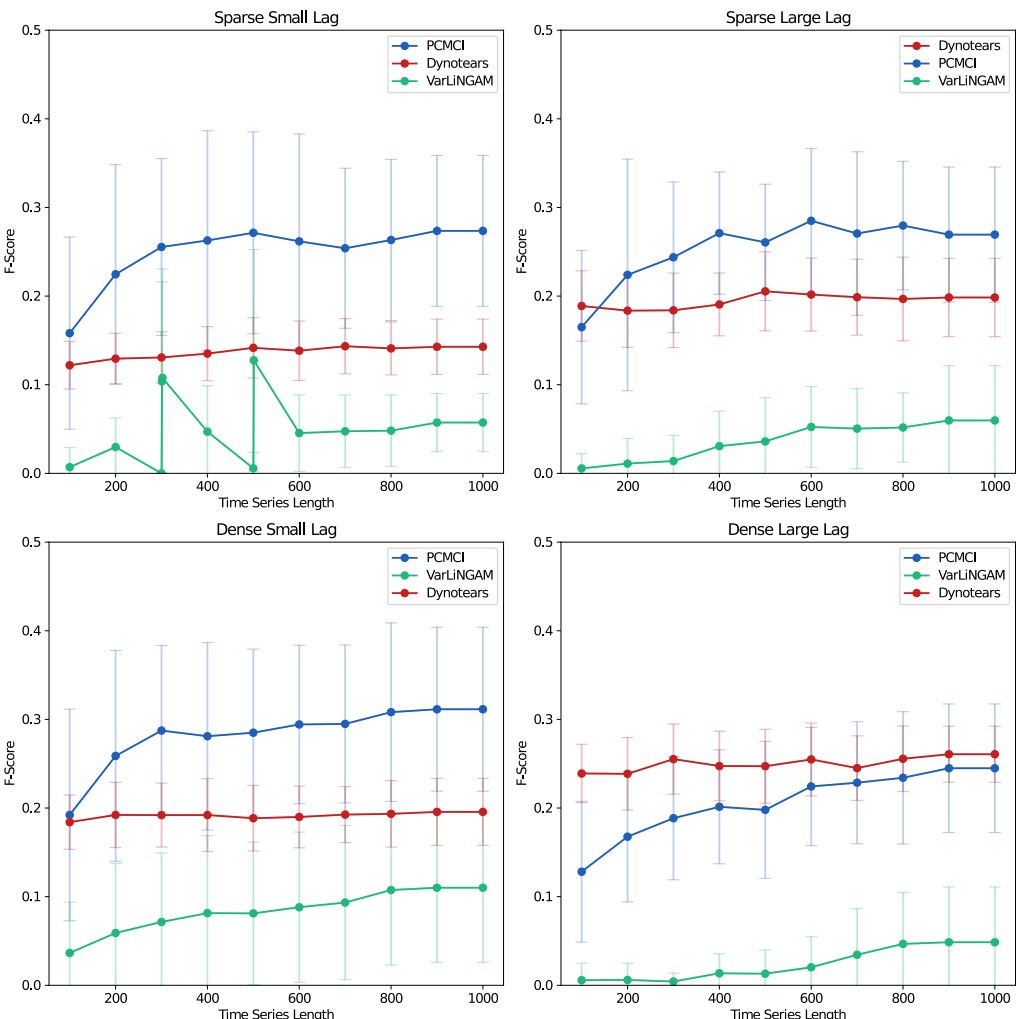

Figure 5: Performance comparison of three causal discovery methods PCMCI, VarLiNGAM, and DYNOTEARS measured by F1-Score across varying time series lengths. The error bars indicates the variances of 5 experiments. Each plot shows the mean F1-Score for three basic structures (star, tree, cycle) under different conditions: sparse small lag (top left), sparse large lag (top right), dense small lag (bottom left), and dense large lag (bottom right). The x-axis represents the time series length, while the y-axis indicates the F1-Score. Error bars depict the standard deviation across the repetitions. The PCMCI method generally outperforms VarLiNGAM and DYNOTEARS in most scenarios, particularly with increasing time series length.

We suggest that this paradigm could potentially be enhanced through the generation of multivariate datasets with configurable spatio-temporal architectures. By allowing users to specify both the graphical model and the functional relationships between variables, we hope to enable the simulation of scenarios that better reflect real-world dynamics. For example, users might define spatial

relationships like network-based or geographical dependencies, along with their temporal evolution patterns.

With the growing adoption of graphical model-based methodologies, there appears to be an increasing need for comprehensive benchmarking frameworks. Our proposed system aims to contribute to this space by providing a platform for evaluating time series forecasting and imputation algorithms using data that attempts to represent spatio-temporal complexities. Through the generation of data with adjustable characteristics, we hope this work may help improve the assessment of algorithms across different scenarios, though further research would be needed to validate its effectiveness.

## 7 CONCLUSION

**Contributions** `Graphical-TS` offers a useful tool for integrating human expertise with algorithmic processes for causal discovery in multivariate time series data. By supporting collaboration between domain experts and computational models, the system enhances the representation of causal dynamics. With its flexible interface, customizable uncertainty features, and the ability to simulate datasets with known ground truth, `Graphical-TS` provides a valuable resource for research in various fields, including physiology. Additionally, the system contributes to the iterative refinement of causal models and the evaluation of causal discovery algorithms, supporting their application in time series data analysis.

**Limitations** The system lacks comprehensive validation of its effectiveness across different user groups and use cases, including medical researchers, data scientists, and domain experts in various fields. More extensive testing is needed to evaluate how different users interact with and benefit from the system's features, particularly in real-world analytical scenarios.

**Future Work** Future research directions include conducting comprehensive user studies with domain experts to validate the system's effectiveness and usability. We plan to enhance visualization capabilities to better represent complex causal relationships and temporal dynamics. Additionally, we aim to incorporate more sophisticated statistical methods for uncertainty estimation and develop robust quantification approaches for learned causal relationships. These improvements will strengthen the system's ability to support reliable causal inference in time series analysis.

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

## A  USER INTERFACE FEATURES

For a demonstration of the interface, please see 2 and 6. Below we've listed some core functionalities of the interface.

**Graph Creation and Editing.** The interface allows users to create causal graphs from scratch or load existing graphs. Users can use the interactive graph editor to add, delete, and modify nodes (representing variables) and edges (representing causal relationships). Nodes and edges can be labeled to reflect their domain-specific meanings. The tool offers an interactive visualization environment that helps users intuitively understand and manipulate the causal structures.

**Integration with Causal Discovery Algorithms.** Graphical-TS integrates with state-of-the-art causal discovery algorithms. Users can utilize these algorithms to generate initial causal graphs from their data. The interface supports iterative refinement, where users modify the algorithmically generated graphs based on domain knowledge and then reapply the algorithms to improve model accuracy.

**Synthetic Data Generation.** The interface provides functionalities for generating synthetic multivariate time series data based on the specified causal graphs. Users can configure the parameters and constraints of the synthetic data generation process to ensure the data accurately reflects the intended causal relationships. This feature is crucial for testing hypotheses and validating causal models.

**Interactive Visualization.** Users can visualize the structure of graphical models interactively and display the corresponding time series data. The visualization tools help users better understand and interpret the causal relationships and data patterns, facilitating deeper analysis.

**Collaborative Environment.** `Graphical-TS` Interface supports interactive editing, allowing multiple users to collaboratively work on the same causal graph. Version control features are included to track changes and revert to previous versions if necessary. This collaborative environment enhances the efficiency of the causal discovery process and facilitates knowledge sharing among research teams.

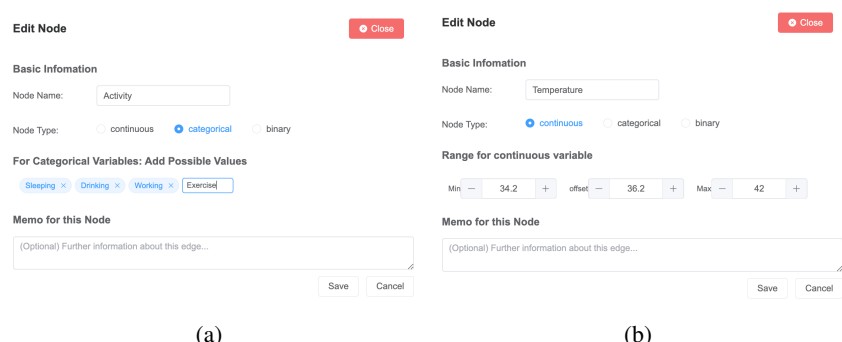

(a)                                (b)

Figure 6: Dialogues for node editing are illustrated using an example from a physiological scenario. Figure **(a)** shows the input interface for adding or editing a categorical variable, such as "activity." The user needs to enumerate the possible values of interest by adding items. In the background, the categories are assigned a unique integer label so the back end can utilize them seamlessly. The memo input can be used to provide additional information for the algorithm developer. Similarly, figure **(b)** shows an example of a continuous variable, "temperature." The minimum and maximum fields define the range, and an offset records a typical value of the variable.

# B  ADDITIONAL EXPERIMENTAL RESULTS

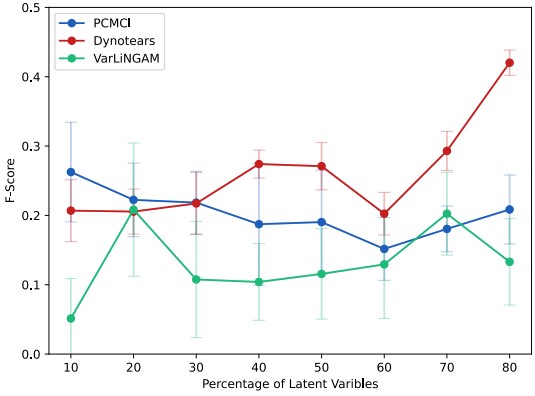

Figure 7: F1-Score comparison of PCMCI, VarLiNGAM, and DYNOTEARS as function of the percentage of latent variables, with the time series length fixed at 600. The chart illustrates the impact of increasing latent variables on the performance of each method in detecting causal relationships in the time series data.

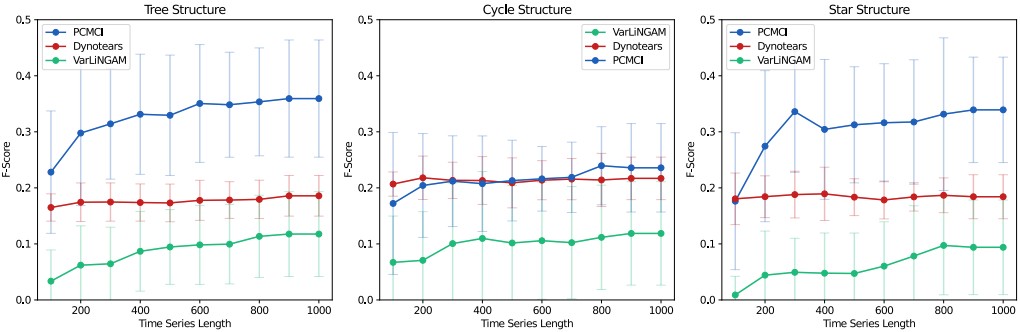

Figure 8: F-Score Results by Structural Configuration: The plots illustrate the performance of three models—PCMCI, Dynotears, and VarLiNGAM—across different time series lengths. Key observations include PCMCI's consistently higher F-scores in the Tree Structure, while Dynotears and PCMCI show similar performance in the Cycle Structure. VarLiNGAM generally exhibits lower F-scores across all structures. Error bars represent variability in the results.

