# OpenReview forum: "Graphical-TS: An Interactive AI Pipeline for Multivariate Time Series with Ground-truth Graphical Modeling"
_ICLR.cc/2025/Conference — ICLR 2025 Conference Withdrawn Submission_

### Official Review · Reviewer_ckh3 · 2024-10-29

**Soundness:** 3
**Presentation:** 4
**Contribution:** 3
**Rating:** 6
**Confidence:** 4

**Summary:**

The paper presents Graphical-TS, an interactive simulation framework for multivariate time series (MTS) incorporating spatiotemporal causal graphical models.

**Strengths:**

- The paper is well written and easy to understand.
- The prior work is clearly described.
- The contributions outlined in Section 4 are valuable for causal graphical modeling, including: synthetic data generation, collaborative environment for interactive causal graph editing, and integration with causal discovery algorithms, among others.
- Figures 2 and 3 effectively illustrate the user interface and provide a useful example of synthetic data generation for variable categories.

**Weaknesses:**

If the authors can address the following points to this reviewer's satisfaction, I would be happy to increase my score.

1. The paper mentions the CausalTime pipeline as prior work, noting that it requires a predefined expert graph and lacks interactive features. The contributions of the proposed interface are clearly outlined in Section 4, but it is unclear whether other interfaces or pipelines include these options as well. A table or summary of similarities and differences between the proposed library and prior work would be helpful for distinguishing contributions. For example, bold paragraphs in Section 4 could serve as table column headers with prior work and proposed methods as row headers, using check marks or X’s to denote the capabilities of each interface/pipeline.

2. An outline of how synthetic data is generated is not provided/clear. A section that outlines the synthetic data generation process/notation and ties it to the example synthetic data generated in Figure 2 would be helpful.

**Questions:**

Questions:
1. Line 189: The paper mentions: “For simplification, we assume the functional f j are additive with respect to Pa(X j t ) and the uncertainty process η”? What is the rationale behind this assumption, aside from simplification? Does prior work or references exist to support this assumption?
2. In Figure 6, the caption states, “Each plot shows the mean F1-Score for three basic structures (star, tree, cycle) under different conditions.” Which model corresponds to which structure? The plot labels three causal discovery methods but does not label the structures.
3. Does the interface support the option for specific types of categorical features, such as nominal and ordinal?

Suggestions:
- Fig. 1: Recommend explaining the various color options of nodes and variables to summarize the information in the figure.
- Line 207: f_j perhaps should b f^j to correspond with prior notation. Would suggest using f^(j) to denote an index j rather than an exponent.

---

> ### Author Response · Authors · 2024-11-28
> **Replies to Weaknesses**
>
> We really appreciate the valuable comments and questions from reviewer ckh3, which are very helpful for our paper improvement. We tried to address all the comments and questions below. The paper has also been updated accordingly in the newest PDF.
> > The paper mentions the CausalTime pipeline as prior work, noting that it requires a predefined expert graph and lacks interactive features. The contributions of the proposed interface are clearly outlined in Section 4, but it is unclear whether other interfaces or pipelines include these options as well. A table or summary of similarities and differences between the proposed library and prior work would be helpful for distinguishing contributions. For example, bold paragraphs in Section 4 could serve as table column headers with prior work and proposed methods as row headers, using check marks or X’s to denote the capabilities of each interface/pipeline.
>
> We appreciate these great suggestions and have modified the paper accordingly in the updated PDF.
>
>
> > An outline of how synthetic data is generated is not provided/clear. A section that outlines the synthetic data generation process/notation and ties it to the example synthetic data generated in Figure 2 would be helpful.
>
> We have added some extra caption to figure 2 as per suggested

---

> ### Author Response · Authors · 2024-11-28
> **Replies to Questions**
>
> > Line 189: The paper mentions: “For simplification, we assume the functional f j are additive with respect to Pa(X j t ) and the uncertainty process η”? What is the rationale behind this assumption, aside from simplification? Does prior work or references exist to support this assumption?
>
> Additive functional relationships are fitting most assumptions of SOTA causal discovery methods on time series such as DYNOTEARS, PCMCI
>
> > In Figure 6, the caption states, “Each plot shows the mean F1-Score for three basic structures (star, tree, cycle) under different conditions.” Which model corresponds to which structure? The plot labels three causal discovery methods but does not label the structures.
>
> We have added the results with respect to the structures. Please check appendix B in the updated paper.
>
> > Does the interface support the option for specific types of categorical features, such as nominal and ordinal?
>
> They will be encoded as integers in the backend and the order is in ascii, but we don't have dedicated interface for ordinal.

---

### Official Review · Reviewer_uQ8i · 2024-11-04

**Soundness:** 3
**Presentation:** 2
**Contribution:** 2
**Rating:** 5
**Confidence:** 4

**Summary:**

The paper presents Graphical-TS, an interactive framework designed for causal discovery in multivariate time series (MTS) data. It addresses the challenges researchers face in generating and analyzing MTS data, particularly the redundancy in data generation efforts and the need for effective integration of expert knowledge. The authors argue that domain experts should define causal relationships, while algorithm developers focus on refining their algorithms.

Key contributions of the paper include:

User Interface Development: The framework features a comprehensive user interface that allows users to create, edit, and visualize causal graphs, facilitating an iterative process of causal discovery and model refinement.

Integration with Causal Discovery Algorithms: Graphical-TS integrates state-of-the-art causal discovery algorithms, enabling users to generate initial causal graphs from data and refine them based on domain knowledge.

Synthetic Data Generation: The interface supports the generation of synthetic MTS data that reflects specified causal relationships, which is crucial for hypothesis testing and model validation.

Collaborative Environment: The framework promotes collaboration among researchers by allowing multiple users to work on the same causal graph, complete with version control features.

Enhanced Causal Modeling: By enabling the simulation of complex spatiotemporal dependencies, Graphical-TS advances the methodology for evaluating time series forecasting and imputation algorithms, thereby improving the precision of algorithmic evaluations.

Overall, Graphical-TS serves as a valuable tool for integrating human expertise with algorithmic processes, enhancing the representation of causal dynamics in various research fields, including healthcare and finance.

**Strengths:**

Originality: The paper presents Graphical-TS, an innovative framework that integrates expert knowledge with causal discovery algorithms for multivariate time series (MTS) analysis. This originality is evident in several dimensions:
New Problem Formulation: The authors address the challenge of effectively incorporating domain expertise into causal modeling, which has been a limitation in traditional causal discovery methods. By emphasizing the role of human experts in defining causal relationships, the paper redefines the approach to causal discovery in time series data.


Creative Combination of Ideas: The framework combines interactive user interfaces with advanced causal discovery algorithms, allowing for an iterative refinement process. This creative integration enhances the usability and applicability of causal discovery methods, making them more accessible to researchers and practitioners.


Application to New Domains: The focus on generating synthetic data that reflects real-world causal relationships opens up new avenues for research in various fields, including healthcare and finance, where accurate causal modeling is crucial.


Quality: The quality of the paper is commendable, characterized by:
Robust Methodology: The authors provide a comprehensive description of the framework's functionalities, including graph creation, editing, and synthetic data generation. The integration with state-of-the-art causal discovery algorithms is well-articulated, demonstrating a solid understanding of the field.


Empirical Validation: While the paper outlines the framework's capabilities, further empirical results and case studies would enhance the quality of the claims made. However, the theoretical foundation and proposed methodologies are sound and well-supported.


Clarity: The paper is generally well-structured and clear, with strengths in:
Writing Style: The writing is coherent and accessible, making complex concepts understandable for a broad audience. The use of figures and examples effectively illustrates key points, aiding in reader comprehension.


Contextualization: The authors successfully situate their work within the existing literature, highlighting the limitations of current methodologies and the need for expert integration. This contextualization enhances the clarity of the paper's contributions.


Significance: The significance of the paper is substantial, as it addresses pressing issues in causal discovery and MTS analysis:
Impact on Research Community: By providing a framework that enhances the representation of causal dynamics and facilitates rigorous evaluations of forecasting algorithms, the paper contributes valuable tools for researchers in various domains. The potential for improved accuracy in causal modeling has far-reaching implications for fields that rely on data-driven decision-making.


Broader Applications: The ability to generate synthetic datasets that reflect real-world complexities is particularly significant in privacy-sensitive areas, allowing researchers to test hypotheses without compromising sensitive information.

**Weaknesses:**

While the paper presents a compelling framework in Graphical-TS, there are several areas where it could be improved to enhance its contributions and effectiveness. Below are specific weaknesses along with constructive and actionable insights for improvement:

1. Empirical Validation and Case Studies
Weakness: The paper lacks comprehensive empirical validation of the Graphical-TS framework. While it outlines the functionalities and theoretical underpinnings, there are limited real-world case studies or experiments demonstrating its effectiveness in practice.

Actionable Insight:

Incorporate Case Studies: The authors should include detailed case studies that showcase the application of Graphical-TS in real-world scenarios, particularly in fields like healthcare or finance. This could involve analyzing existing datasets to illustrate how the framework improves causal discovery compared to traditional methods.

Benchmarking Against Existing Methods: Conduct systematic comparisons with established causal discovery methods (e.g., PCMCI, GES) using standardized datasets. Presenting quantitative metrics (e.g., precision, recall, F1-score) would provide a clearer picture of the framework's performance and its advantages.

2. User Interface and Usability

Weakness: While the paper mentions an intuitive user interface, it does not provide sufficient detail on how users interact with the system or the specific features that facilitate collaboration between domain experts and data scientists.

Actionable Insight:

Detailed User Interface Description: Include screenshots or diagrams of the user interface to illustrate how users can manipulate causal relationships and input expert knowledge. A walkthrough of the user experience would help readers understand the practical implications of the framework.
User Feedback and Iteration: Consider conducting user studies or surveys with domain experts to gather feedback on the interface and usability. This could inform iterative improvements and ensure that the system meets the needs of its intended users.

3. Limitations of Synthetic Data Generation

Weakness: The paper discusses the generation of synthetic data but does not adequately address the limitations and potential biases that may arise from this approach. Relying solely on synthetic data could lead to overfitting or misrepresentation of real-world dynamics.

Actionable Insight:

Discuss Limitations: The authors should explicitly discuss the limitations of synthetic data generation, including potential biases and the risk of not capturing the full complexity of real-world systems. This could involve a section dedicated to the challenges and considerations when using synthetic data.
Hybrid Approaches: Explore the possibility of integrating real-world data with synthetic data to create a more robust dataset. This could involve using real data to inform the parameters of the synthetic data generation process, ensuring that the generated data reflects realistic scenarios.

4. Theoretical Foundations and Justifications

Weakness: The theoretical foundations of the framework could be more robustly articulated. While the paper references existing methodologies, it does not sufficiently justify the choices made in the design of Graphical-TS.

Actionable Insight:

Theoretical Justification: Provide a more in-depth discussion of the theoretical principles underlying the framework, including why certain algorithms or approaches were chosen over others. This could involve citing relevant literature that supports these choices and discussing their implications for the framework's performance.

Addressing Potential Critiques: Anticipate and address potential critiques of the framework, such as concerns about scalability or the handling of high-dimensional data. Discuss how Graphical-TS can be adapted or improved to address these challenges.

**Questions:**

Empirical Validation of Graphical-TS

Question: Can you provide specific examples of real-world datasets where Graphical-TS has been applied? What were the outcomes of these applications?

Suggestion: Including empirical results from case studies would strengthen the paper. Consider adding a section that details the application of Graphical-TS in a specific domain, such as healthcare or finance, along with quantitative metrics that demonstrate its effectiveness compared to existing methods.

Limitations of Synthetic Data

Question: What measures have been taken to ensure that the synthetic data generated by Graphical-TS accurately reflects real-world complexities? How do you address potential biases in this data?

Suggestion: A discussion on the limitations of synthetic data generation and how it may impact the validity of the causal models would be valuable. Consider exploring hybrid approaches that combine real-world data with synthetic data to enhance robustness.

Theoretical Foundations

Question: Can you elaborate on the theoretical principles that guided the design of Graphical-TS? Why were certain algorithms or methodologies chosen over others?

Suggestion: Providing a more detailed theoretical justification for the framework's design choices would enhance its credibility. Including references to relevant literature that supports these choices could also strengthen the argument.

Scalability and High-Dimensional Data

Question: How does Graphical-TS handle scalability issues, particularly when dealing with high-dimensional datasets? Are there any limitations in this regard?

Suggestion: Addressing potential scalability challenges and discussing how the framework can be adapted to handle high-dimensional data would be important. Consider including performance benchmarks or examples that illustrate how the system performs under varying data conditions.

Integration of Expert Knowledge

Question: How is expert knowledge integrated into the causal discovery process? What mechanisms are in place to ensure that this knowledge is accurately represented in the graphical models?

Suggestion: A clearer explanation of how expert input is incorporated and refined within the framework would be beneficial. Discussing the iterative process of integrating human insights with algorithmic learning could provide a more comprehensive understanding of the system's functionality.

Future Directions and Improvements

Question: What are the planned future enhancements for Graphical-TS? Are there specific features or functionalities that you aim to develop based on user feedback?

Suggestion: Outlining future directions for the framework, including potential improvements or new features, would provide insight into the authors' vision for the system. This could also encourage collaboration and engagement from the research community.

---

> ### Author Response · Authors · 2024-11-27
> **Replies to Weaknesses**
>
> We appreciate your valuable feedback. We will reply to each of your comment.
>
> 1. **Limitations on User Studies:**
> We acknowledge the current limitations regarding user studies. However, competitive benchmarking against other casual discovery method is not applicable to us because we are providing the dataset for benchmarking.
>
> 2. **Simulation Framework and User Interface:**
> Our primary focus is to present the simulation framework, with the user interface serving as a demonstration of the APIs. We plan to provide a comprehensive manual for the interface outside of the paper. While many functionalities will be explored in future user studies, as suggested by other reviewers, they are not the main focus at this time.
>
> 3. **Expansion of Limitations and Data Fusion:**
> i. We will expand on the limitations in future iterations of the paper.
> ii. We are considering whether to address the absence of real-data fusion, which is due to the current focus of the paper.
>
> 4. **Theoretical Foundation:**
> We would appreciate specific feedback on the introduction of the theoretical foundation so that we can consider how best to revise it.

---

> ### Author Response · Authors · 2024-11-28
> **Replies to Questions**
>
> > Empirical Validation of Graphical-TS
> > Question: Can you provide specific examples of real-world datasets where Graphical-TS has been applied? What were the outcomes of these applications?
>
> This is indeed a limitation of the current stage of work. User study will be an major direction to extend our paper in the future. Thereby we added a statement about this in the conclusion part.
>
> > Limitations of Synthetic Data
> > Question: What measures have been taken to ensure that the synthetic data generated by Graphical-TS accurately reflects real-world complexities? How do you address potential biases in this data?
> > Suggestion: A discussion on the limitations of synthetic data generation and how it may impact the validity of the causal models would be valuable. Consider exploring hybrid approaches that combine real-world data with synthetic data to enhance robustness.
>
> We can, distributional-wise, incorporate real-world dataset to the simulated data with measure like KL divergence, but we need to trade off the customizability and the distributional consistency with the real-world data. This requires careful study and the aforementioned user study seems to be a prerequisite.
>
> > Theoretical Foundations
> > Question: Can you elaborate on the theoretical principles that guided the design of Graphical-TS? Why were certain algorithms or methodologies chosen over others?
> > Suggestion: Providing a more detailed theoretical justification for the framework's design choices would enhance its credibility. Including references to relevant literature that supports these choices could also strengthen the argument.
>
>
>
> > Scalability and High-Dimensional Data
> > Question: How does Graphical-TS handle scalability issues, particularly when dealing with high-dimensional datasets? Are there any limitations in this regard?
> > Suggestion: Addressing potential scalability challenges and discussing how the framework can be adapted to handle high-dimensional data would be important. Consider including performance benchmarks or examples that illustrate how the system performs under varying data conditions.
>
> We agree that we should be aware of the scalability. In the future this is another meaningful take to enrich the experimental content.
>
> > Integration of Expert Knowledge
> > Question: How is expert knowledge integrated into the causal discovery process? What mechanisms are in place to ensure that this knowledge is accurately represented in the graphical models?
> > Suggestion: A clearer explanation of how expert input is incorporated and refined within the framework would be beneficial. Discussing the iterative process of integrating human insights with algorithmic learning could provide a more comprehensive understanding of the system's functionality.
>
> Our system recognizes that expert knowledge may carry biases and quantifies this with an uncertainty measure. This uncertainty can be mitigated through collaboration and the integration of insights from multiple experts. Additionally, the system is designed to function as a component of a human-in-the-loop pipeline, incorporating both human expertise and statistical or algorithmic results.
>
>
>
> > Future Directions and Improvements
> > Question: What are the planned future enhancements for Graphical-TS? Are there specific features or functionalities that you aim to develop based on user feedback?
> > Suggestion: Outlining future directions for the framework, including potential improvements or new features, would provide insight into the authors' vision for the system. This could also encourage collaboration and engagement from the research community.
>
> We have included a new paragraph about the future work.

---

### Official Review · Reviewer_sggw · 2024-11-04

**Soundness:** 2
**Presentation:** 1
**Contribution:** 2
**Rating:** 3
**Confidence:** 3

**Summary:**

The paper presents an interactive framework for time series generation that incorporates causal graphs. It offers the implementation details and a UI description and compares the proposed method to other structure learning methods for time series.

**Strengths:**

- **An important research problem**. I believe that such a system is instrumental for many applications, such as healthcare,
- **The practical applicability of the developed system**. The paper develops a tool that considers domain experts and simplifies structure learning in time series data.

**Weaknesses:**

Here, I will summarize the weaknesses; see Questions for more detailed comments and suggestions.

- Writing. In some parts of the paper, the results do not support the claims, and in some places, they are unclear. Overall, it looks like the text requires additional significant polishing,
- The experimental design. The paper claims multiple times the benefit of the proposed system to domain experts. It seems crucial to support it with a user-study,
- Fairness evaluation. When a system considers input from domain experts, it becomes prone to biases and might have fairness issues. It is important to discuss those in this work clearly.
- The relevance to ICLR. The paper's main contribution is the developed user interface and the system. Though, in my opinion, it is more a focus of HCI conferences rather than ICLR,

**Questions:**

> an uncertainty process

It is not a conventional term, but it is worth expanding.

> LL143-144: the definition of Pa:

It seems that it takes into account all the data points from previous steps, but should it not also take into account the causal parent from timestep t+1?

[Writing]: I think it is worth separating the implementation details from the methodological contributions. For instance, in Sec.3.3, they are mixed.


[Sec 3.2.]: it is worth discussing why the assumptions are realistic

>A common practice in causal discovery research is to generate Erd˝os–R´enyi graphs

Please add references

[Fig 6]: fonts are too small; how err bars are calculated?

[Sec. 6]: I think the claims of this section should be supported with concrete experiments.

> . By supporting collaboration between domain ex-
parts and computational models

Are there any use studies to support collaboration with domain experts?

**Details Of Ethics Concerns:**

The paper proposes a system that uses domain experts to generate time series data. It is essential to ensure that the input from domain experts does not cause biases or fairness issues in the generated data.

---

> ### Author Response · Authors · 2024-11-27
> **Replies to Weaknesses**
>
> > _"Writing..."_
>
> We appreciate your feedback and would love to hear more about the specific areas that caught your attention.
>
> > _"The experimental design..."_
>
> The enhancement of the experimental design is a planned extension of this study and will be addressed in future work. We have included a statement regarding this in the limitations and outlook section of the paper.
>
> > *"Fairness evaluation... "*
>
> Our system recognizes that expert knowledge may carry biases and quantifies this with an uncertainty measure. This uncertainty can be mitigated through collaboration and the integration of insights from multiple experts. Additionally, the system is designed to function as a component of a human-in-the-loop pipeline, incorporating both human expertise and statistical or algorithmic results.
>
> > *"The relevance to ICLR..."*
>
> Although the paper could be re-positioned as an HCI system paper, highlighting our interface, our goal for the conference is to introduce it as a benchmarking system with strong agility.

---

> ### Author Response · Authors · 2024-11-27
> **Replies to Questions**
>
> >*"an uncertainty process"*
>
> The conventional expression seems slightly simple for our settings. We seek to model the uncertainty more carefully in the future therefore we think it is a good idea to look at it as a separate process.
>
>
> > *"LL143-144: the definition of Pa:..."*
>
> Thank you for pointing it out; we have addressed that by removing the confusing notations
>
> > *"I think it is worth separating the implementation..."*
>
> We agree with this suggestion totally and have revised accordingly.
>
> > *"it is worth discussing why the assumptions are realistic"*
>
> We have briefly elaborated on the advantages of using the additive assumption.
>
>
> > reference needed : *"A common practice in causal discovery research is to generate Erd˝os–R´enyi graphs"*
> > *"fonts are too small; how err bars are calculated?"*
>
> Please see the revision
>
> > *"I think the claims of this section should be supported with concrete experiments. .."*
>
> We want to highlight first the application to causal discovery, so we decided to not show the experiments for forecasting and imputation. Another reason is that it is interesting when can compared forecasting and imputation performances with vs. without the underlying graphical model; however algorithms for the latter case is not very common. But in general this is a nice suggestion, so we change the tone from a claiming style to a more discussing one.
>
> > *"Are there any use studies to support collaboration with domain experts?"*
>
> Unfortunately this is a major limitation of this paper. We have included this in the conclusion.

---

### Official Review · Reviewer_Hst4 · 2024-11-11

**Soundness:** 2
**Presentation:** 3
**Contribution:** 1
**Rating:** 3
**Confidence:** 3

**Summary:**

The paper describes a system that generates synthetic multivariate time series data with known causal relationships to evaluate causal discovery algorithm.

**Strengths:**

* originality: the paper describes a system to generate synthetic data that can be used for causal discovery algorithms. The data generation process is based on Runge, 2020. The originality of the paper is fairly limited.
* quality: the system supports features such as expert knowledge integration. The data generation process is also reasonable.
* clarity: the paper is clearly written.
* Significance: the paper is based on established algorithmic techniques and hence has limited algorithmic significance.

**Weaknesses:**

my major concerns about the paper are its originality and significance. It seems that the major contribution of the paper is to implement existing data generation techniques. I do not see the contributions made in the paper is sufficient to justify its publication in this venue.

**Questions:**

The author can consider clarifying the significance and originality of the paper. Real-world use cases beyond standard experiments can also be helpful.

---

> ### Author Response · Authors · 2024-11-25
>
> Thank you for your valuable feedback. We acknowledge the concern regarding the originality and significance of our work. Our primary objective in this paper is to introduce the agility of our simulation framework and its potential as a general benchmarking system. We understand that not all functionalities were described in a mathematical way, such as the definition of graph perturbation, one of the core feature of our framework. We will consider adjusting the focus of our presentation to better highlight these aspects and provide a more comprehensive mathematical description in future revisions.

---

### Note · Authors · 2025-01-22

I have read and agree with the venue's withdrawal policy on behalf of myself and my co-authors.